# Recognition of the Airspace Affected by the Presence of Volcanic Ash from Popocatepetl Volcano Using Historical Satellite Images

**José Carlos Jiménez-Escalona** [1,*] **, José Luis Poom-Medina** [2] **, Julie Roberge** [3] **, Ramon S. Aparicio-García** [1] **, José Eduardo Avila-Razo** [1] **, Oliver Marcel Huerta-Chavez** [1] **and Rodrigo Florencio Da Silva** [1]

1    Escuela Superior de Ingeniería Mecánica y Eléctrica, Instituto Politécnico Nacional, Av. Ticomán 600, Col. San José Ticomán, Mexico City 07340, Mexico; rapariciog@ipn.mx (R.S.A.-G.); jeavilar@ipn.mx (J.E.A.-R.); ohuertac@ipn.mx (O.M.H.-C.); rflorencio@ipn.mx (R.F.D.S.)
2    Physics Research Department, Universidad de Sonora, Rosales y Boulevard Luis Encinas S/N, Hermosillo 83000, Mexico; joseluis.poom@unison.mx
3    Escuela Superior de Ingeniería y Arquitectura, Instituto Politécnico Nacional, Av. Ticomán 600, Col. San José Ticomán, Mexico City 07340, Mexico; jroberge@ipn.mx
*    Correspondence: jjimeneze@ipn.mx

**Abstract:** A volcanic eruption can produce large ash clouds in the atmosphere around a volcano, affecting commercial aviation use of the airspace around the volcano. Encountering these ash clouds can cause severe damage to different parts of the aircraft, mainly the engines. This work seeks to contribute to the development of methods for observing the dispersion of volcanic ash and to complement computational methods that are currently used for the prediction of ash dispersion. The method presented here is based on the frequency of occurrence of the regions of airspace areas affected by ash emission during a volcanic eruption. Popocatepetl volcano, 60 km east of Mexico City is taken as a case study. A temporal wind analysis was carried out at different atmospheric levels, to identify the direction towards which the wind disperses ash at different times of the year. This information showed two different trends, related to seasons in the direction of dispersion: the first from November to May and the second from July to September. To identify the ash cloud and estimate its area, a set of 920 MODIS images that recorded Popocatepetl volcanic activity between 2000 and 2021 was used. These satellite images were subjected to a semi-automatic, digital pre-processing of binarization by thresholds, according to the level of the brightness temperature difference between band 31 (11 μm) and band 32 (12 μm), followed by manual evaluation of each binarized image. With the information obtained by the processing of the MODIS image, an information table was built with the geographical position of each pixel characterized by the presence of ash for each event. With these data, the areas around Popocatepetl volcano with the highest frequency of affectation by ash emissions were identified during the period analyzed. This study seeks to complement the results obtained by numerical models that make forecasts of ash dispersions and that are very important for the prevention of air navigation risks.

**Keywords:** volcanic monitoring; satellite images; aviation risk; hazard mitigation; air navigation

## 1. Introduction

Aircraft encounter with volcanic ash is a very important issue because of its risk implication for civil aviation safety [1,2]. Continuous volcanic eruptions of low to medium intensity (Volcanic Explosivity Index [VEI] 1 to 3) create a high risk of an ash cloud for aviation using the airspace around active volcanoes. The presence of volcanic ash at jet-cruise atmospheric levels compromises the safety of aircraft operations and forces re-routing of aircraft to prevent encounters with volcanic ash clouds. In some cases, the airports even need to close around the affected zone. In the past decades, volcanic crises (e.g., Chaitén

2008, Cordón Caulle 2011 and Calbuco 2015, Chile; Eyjafjallajökull 2010, Iceland) clearly demonstrated that low to violent (VEI 1 to 5) eruptions, particularly if long-lasting, can paralyze entire sectors of societies with a significant economic impact [3–7]. The eruption of Eyjafjallajokull volcano in Iceland (2010) provided the perfect study case to analyze the strong economic impact of volcanic eruptions on aviation. Between 14 and 21 April 2010, more than 100,000 flights were cancelled, with more than US$1.7 billion in lost revenues for airlines and more than 10 million stranded passengers [8]. The hazard can extend for several tens of kilometers in the prevailing wind direction [9]. Other recent eruptions that affected flights and airport closures were from the active volcanoes Mount Agung in Bali (2017), Raikoke volcano in Russia (2019) [10], and Mount Sinabung on the Island of Sumatra (2019) in Indonesia [6]. In addition, in January 2020 [11], Mount Taal volcano in the Philippines caused the closure of the airport in Manila resulting in the cancellation of more than 500 flights [6].

Several tools have been developed, based on the monitoring and tracking of ash clouds through satellite images [12–17]. Furthermore, significant advances have also been made in the development of ash dispersion models focused on mitigating aviation risks (e.g., [18–22]). However, forecast accuracies are limited by poor constraints on eruption source parameters, including how high the ash is emplaced at the source, the mass eruption rate, and the near-source plume dynamics [23]. Another factor of great importance is the meteorological database used, among them the wind field, which has a strong influence on the dimensions of the area that can be affected by the dispersion of the ash cloud [24]. It has been observed that these dispersion models are based mainly on forecasts of meteorological conditions, specifically on the possible changes that the wind could present at the different levels of the atmosphere [25]. This dependence makes this type of ash dispersion model present good results in short-term forecasts (up to 18 h) but causes an increase in the uncertainty of the results when used to make forecasts for longer periods of time [26].

The volcanic ash advisory centers (VAACs) are responsible for producing volcanic ash cloud analysis and forecasts to assist the aviation community. There are currently nine VAACs that provide a comprehensive global modeling and warning system for the aviation operation. VAACs use six different volcanic ash transport and dispersion (VATD) models to produce volcanic ash charts showing the forecast location of volcanic ash in the atmosphere at different flight levels and to forecast lead times of 24 h [27].

Popocatepetl volcano (19.02° N, 98.62° W, 5425 masl), which was taken as a case study, has continuously exhibited periods of explosive activity, producing ash emissions since its reawakening in December 1994. This active volcano located 60 km from Mexico City, is surrounded by important international airports, as shown in Table 1. This implies that, in the airspace used for aircraft traffic, there is also an important set of airways in this area, as can be seen in Figure 1.

**Table 1.** Airports around the Popocatepetl volcano within a radius of 110 km.

| Airport Name | ICAO Code (2005) | Distance from Popocatepetl (km) |
| --- | --- | --- |
| Puebla International Airport | MMPB | 31 |
| Mexico City International Airport | MMMX | 65 |
| Cuernavaca International Airport | MMCB | 82 |
| Atizapan airport | MMJC | 95 |
| Toluca International Airport | MMTO | 106 |

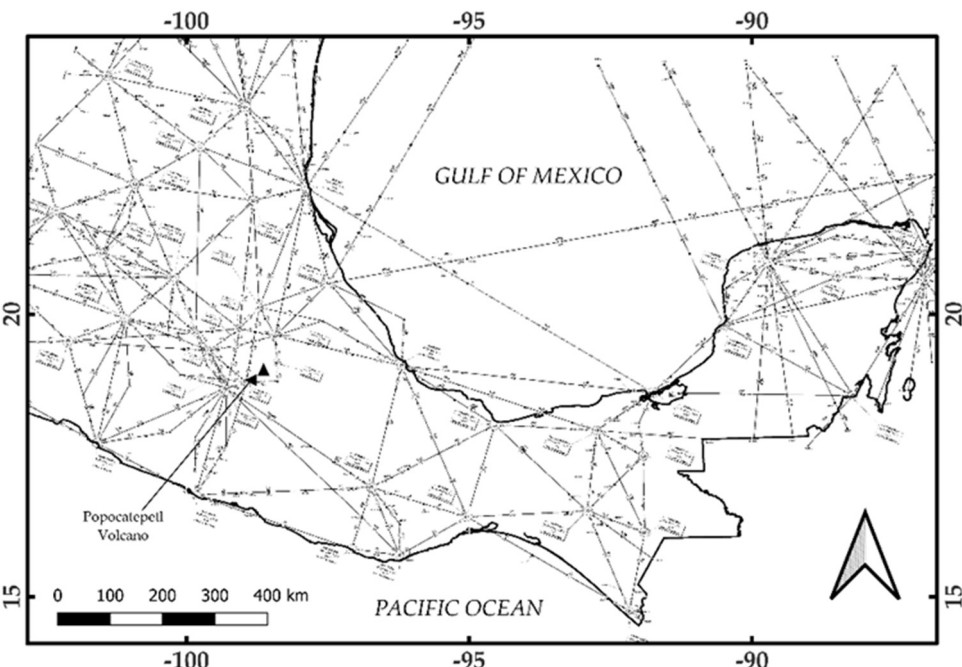

**Figure 1.** Airway map of Central Mexico. The Popocatepetl volcano is located in the central part of the country. (Modified from aeronautical chart of the upper airspace L2, published by SENEAM (Servicio a la Navegación en el Espacio Aéreo Mexicano, in Spanish)).

This aeronautical sector risk is continually increasing due to the rapid expansion of air traffic around the world [28], increasing the probability of an aircraft encountering an ash cloud around active volcano zones.

This paper provides an observational method based on the temporal analysis of satellite images and analysis of dominant patterns of seasonal winds in the upper levels of the troposphere, to identify airspace regions with a high probability of being affected by the presence of volcanic ash around Popocatepetl volcano. This allowed identification of the areas of airspace with the greatest probability of being affected at different times of the year, enabling alerts of the possibly affected areas, that could be confirmed later with the other tools that have been developed.

## 2. The Impact of Volcanic Ash on Aviation

Volcanic ash emission is a hazard to aviation because the ash particles, smaller than 100 microns in diameter, can be transported several hundreds of kilometers from the volcano's crater. This fine ash can be absorbed by a plane's air intakes, accumulate in filters, and even reach electrical and mechanical parts of the aircraft, which can affect its operation [29]. Furthermore, the ash suspended in the atmosphere can be ingested by jet engines, causing erosion in the engine's first stages. When passing through the combustion chamber, the ash can melt, changing its structure to a ductile material that can adhere to the combustion chambers' walls, mainly in the first stages of the turbine. This adhesion effect obstructs the cooling ducts of the turbine blades, causing overheating, which leads to further physical damage and severe in-flight problems [2].

A commercial aircraft travels at speeds of ~800 km/h and at altitudes ranging from 6–14 km. Additionally, to travel from one airport to another, a flight plan authorized by the country's authorities must be followed and the air traffic offices informed, where the route to follow is traced. A defined number of control points, where the aircraft must make an arrival report, identifies each airway. These checkpoints make an air route mandatory, can only be changed by the pilot in case of an emergency, and must be authorized by the air traffic controller.

Based on market studies, the AIRBUS Company forecasts an annual growth of air traffic of 4.6% in the next 20 years [28]. Currently, the global airline fleet number is ~23,600 aircraft, including both passenger and cargo aircraft. Forecast projections indicate that the number of aircraft flying in a specific airspace will double by 2035. This increase in aircraft traffic also increases the probability of encounters with volcanic ash clouds in active volcanic regions. Guffanti et al. [2] observed these effects, reporting a total of 129 aircraft encounters with volcanic ash between 1953 and 2009. Christmann et al. [30] reported 113 aircraft encounters with volcanic ash between 2010 and 2021. Of these 113 aircraft incidents, 92 were attributed to the Eyjafjallajökull volcano's eruption in April 2010. These studies show the need to develop tools that can help to monitor volcanic ash emissions to prevent and mitigate aviation risks due to these volcanic phenomena.

## 3. Study Area and Volcanic Activity

The area of the airspace where the aircraft circulate is located between 3 and 15 km in altitude. Airways at these atmospheric levels must be used by aircraft to move from one airport to another. The airways are designed in such a way as to avoid the areas of greatest danger; however, they can become contaminated by volcanic ash when it is carried by the wind, several hundred kilometers away from where it was emitted, generating a risk for air navigation. Figure 1 shows around 60 airways in upper airspace, in an area covering 370 km (200 Nautical Milles) radius around the volcano crater. The specialized website www.flightradar24.com (accessed on 16 February 2022) indicates a constant average of 70 aircraft flying the airways around the Popocatepetl volcano.

Since Popocatepetl's reawakening in 1994, Mexico's National Center for Disaster Prevention (Centro Nacional de Prevención de Desastres, or CENAPRED) reported a continuous increase in explosive activity within the studied period (Figure 2), reaching a peak in 1999 and 2000, where the Washington VAAC reported almost 120 explosive eruptions each year. After this peak, there was a decrease in activity, leaving less than 20 eruptions per year from 2004 to 2011. However, from 2012 there was a new increase in the volcano's eruptive activity, reaching almost 60 registered eruptions during the years 2012 and 2013, with a maximum number of events in 2019 (262 eruptions) and 2020 (286 eruptions) as can be seen in Figure 2. All these eruptions were cataloged as low to medium intensity explosive events (VEI 1, VEI 2 and a few VEI 3 explosions) by Washington VAAC, and threw out thousands of tons of solid material in particle form (with diameters from several centimeters to several tens of microns). Owing to its weight, the largest sized material is deposited a few kilometers from the crater, depending on the energy expelled to the atmosphere. However, the smallest sized material ($\leq 100$ µm), was transported by the prevailing wind reaching the atmosphere (between 6 and 11 km altitude) and traveling several hundred kilometers before being dispersed by the wind and deposited on the surface of the Earth. During this eruptive period Popocatepetl's ash fall was frequently reported at the Puebla International Airport and at least twice at the Mexico City International Airport (MCIA)including at least four aircraft encounters [2].

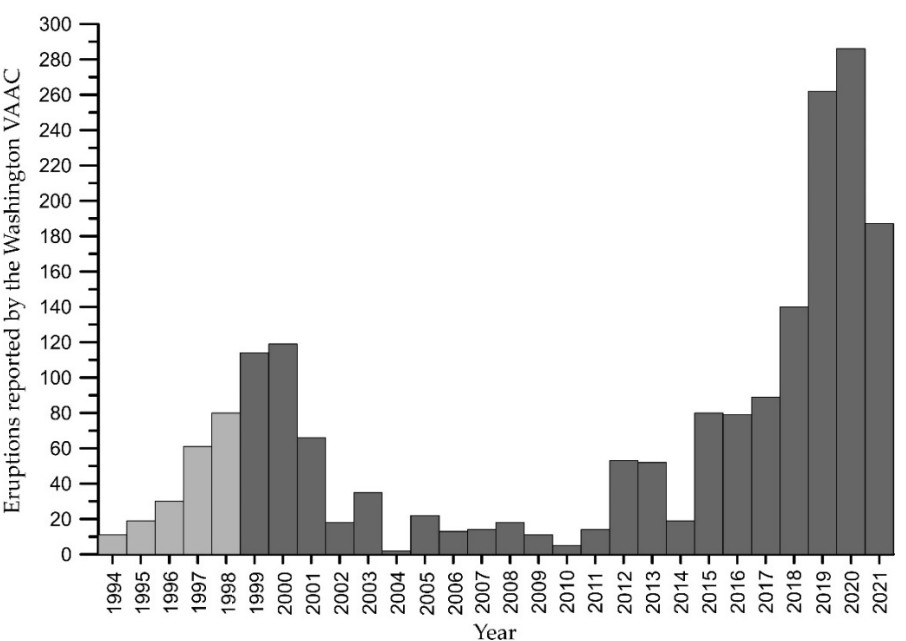

**Figure 2.** Historical activity of the last eruptive period of the Popocatepetl volcano. The information was obtained from CENAPRED reported for the period 1994 to 1998 (light grey columns), and from Washington VAAC, reported for the period 1999 to 2021 (dark grey columns).

## 4. Methodology

To identify the areas with the highest probability of the presence of volcanic ash in the case of an eruption of type VEI 1 to 3, such as those that are periodically produced by the Popocatepetl volcano, 8 steps were carried out (Figure 3).

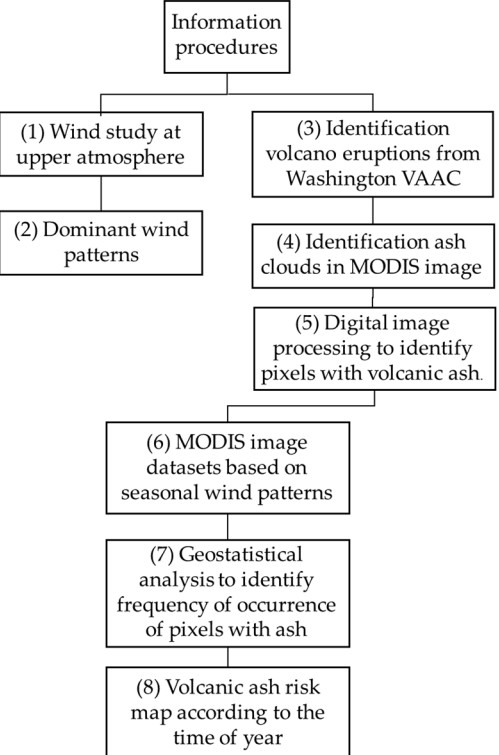

**Figure 3.** Procedure performed for the identification of explosive events in the study period and the digital processing of satellite images to identify areas affected most frequently by volcanic ash.

(1) A characterization study of wind conditions was carried out at different levels of the upper troposphere, with the purpose of (2) identifying the dominant patterns of wind direction throughout the year.

(3) A database was generated, including the date and time, of all Popocatepetl eruptions reported by the Washington VAAC in the period from 2000 to 2021.

(4) Each of the MODIS images that relatively coincided with the time of eruption was searched and the brightness temperature difference (BTD) technique was applied to identify the geographical location and the coverage area of the ash cloud.

(5) To extract the information on the number of pixels identified with the presence of volcanic ash and their geographical location in each of the MODIS images analyzed, a data thresholding technique was used considering only pixels with negative values. Then, morphological close operations were applied in order to delimit the contour of the ash cloud area. The thresholding technique was used after dropping off to zero the pixels with positive values, keeping the negative values as possible ash pixels according to the split window technique. From the set of these negative values corresponding to each pixel, the mean of the distribution frequency is obtained, which was used in each image as the discrimination threshold. Then, the values below the mean threshold were taken as ash pixels.

After the digitalization the images were transformed to matrix form, we could realize using morphological theory to image processing, where the dilate and closing morphological operations were necessary only to consolidate the binarized masks, while the dilate kernel operator size was set to 4 pixels, and 8 pixels for closing.

(6) With this information, two data sets were generated from the digital processing of MODIS images based on seasonal wind patterns found in the wind analysis.

(7) It was assumed that each geographical location point within each of the images could be identified with the pixel to which these coordinates corresponded, so to calculate the frequency with which this geographical location was identified with the presence of ash, a geostatistical calculation was carried out to obtain the frequency of appearance of each pixel in the set of MODIS images obtained in the analysis period based on the two periods of the year defined from the wind study.

(8) As a result, this generated an information matrix that identified the areas of the airspace that had the highest percentage of volcanic ash occurrence in the explosive events of Popocatepetl volcano during the studied period, considering the wind direction patterns in the upper part of the troposphere.

## 5. Data Analysis

### 5.1. Wind Analysis

An essential factor that must be considered when monitoring and predicting volcanic ash dispersion is the atmospheric condition. In recent decades, several modeling tools have been developed to reproduce and predict volcanic ash dispersion [18,31,32]. In addition, Prata [33] provide a good description of remote sensing techniques developed over the last 40 years for detection of volcanic ash and $SO_2$ using satellite information. These techniques have been applied by the nine VAACs and have supported aviation worldwide.

The volcanic ash emitted into the atmosphere in an explosive volcanic event reaches a maximum height that is a function of vent radius, gas exit velocity, gas content of eruption products, and efficiency of conversion of thermal energy contained in juvenile material to potential and kinetic energy during the entrainment of atmospheric air [34]. Once the volcanic products are emitted, they rise until they are thermally balanced with the atmospheric air around them. From this zone of accumulation of stability with the atmosphere, the ash cloud is transported by the dominant wind. The height of this region cannot be known until the eruption occurs. For this reason, in this work, wind profiles at different altitude levels, based on pressure heights, were used. The wind profiles in the atmosphere's vertical structure over the volcano crater were obtained from the web-based Real-time Environmental Applications and Display sYstem (READY) [35], using the NCEP/NCAR (National Center for Environmental Prediction/National Center for

Atmospheric Research) Reanalysis 1data from NOAA (National Oceanic and Atmospheric Administration) 4-times daily in the period between 2000 and 2020. The pressure levels were analyzed from 500 to 300 mb, as shown in Figure 4. The pressure altitudes or flight levels (FL) used by aircraft, which are a standardized heights in hundreds of feet relative to the international standard atmospheric pressure, are shown in Table 2, corresponding to the upper airspace characterized by the International Civil Aviation Organization (ICAO), the area where most air transport operations occur.

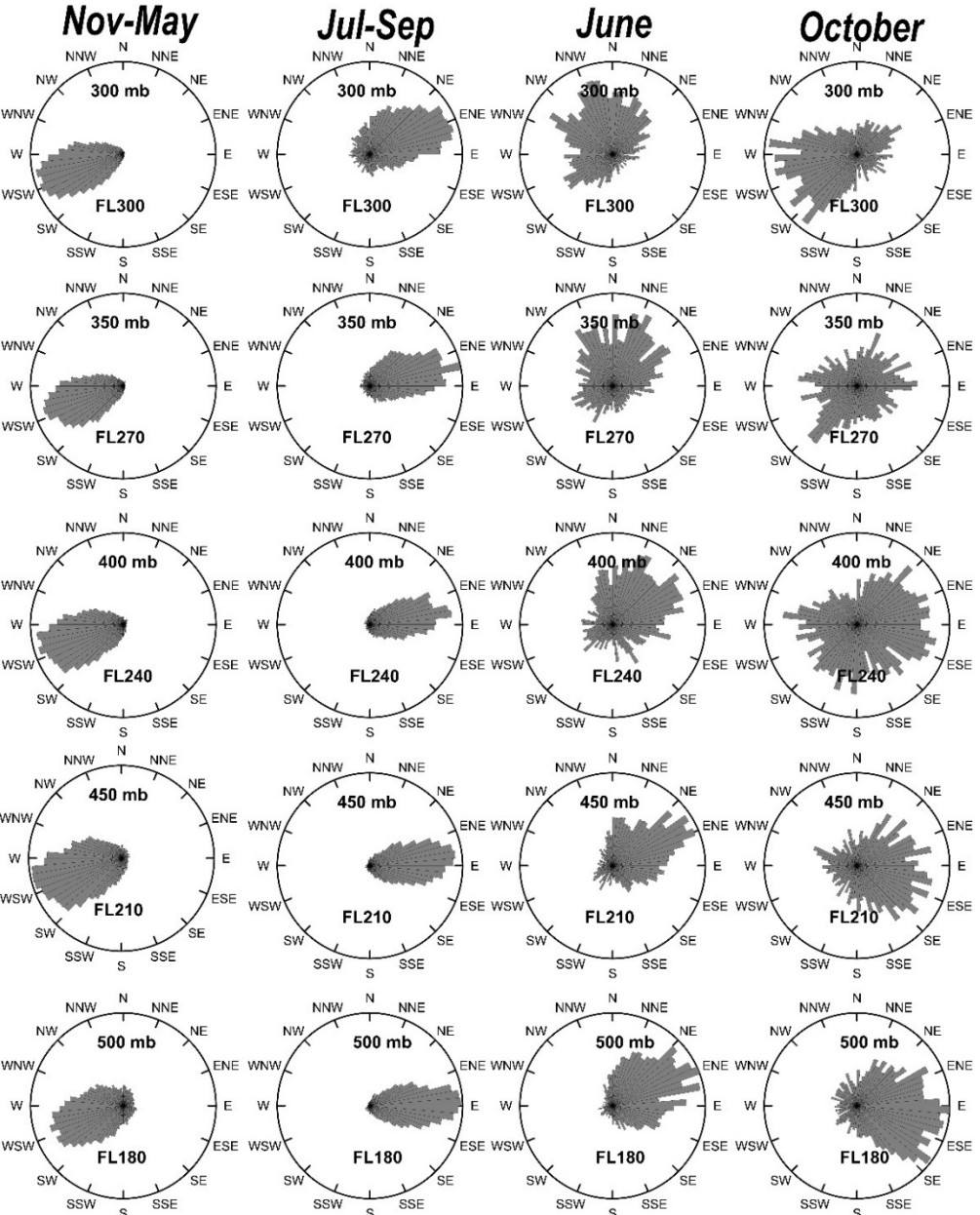

**Figure 4.** Wind rose diagrams over the crater of the Popocatepetl volcano. Data were obtained from NOAA re-analysis data for a period of 20 years, at pressure levels of 500 to 300 mb. Wind data were statistically analyzed separately for each month of the year, associating the trends in the months of the year.

**Table 2.** Flight levels in the airspace and its corresponding pressure level.

| Pressure Level | Flight Level (FL) | Altitude (ft) | Altitude (masl) | Height above Crater (m) |
|---|---|---|---|---|
| 300 mb | FL300 | 30,000 | 9200 | 3775 |
| 350 mb | FL270 | 27,000 | 8200 | 2775 |
| 400 mb | FL240 | 24,000 | 7300 | 1875 |
| 450 mb | FL210 | 21,000 | 6400 | 975 |
| 500 mb | FL180 | 18,000 | 5500 | 75 |

Wind data were statistically analyzed using a wind rose or circular histograms, with tertiary intercardinal directions (32 directions). The data were separated by months of the year, to identify seasonal patterns. This study allowed us to identify two trend patterns in the wind's predominant direction, as observed in Figure 4. The first trend was observed between November to May, whose predominant wind direction is from WSW; and the second dominant trend occurs between July to September, whose predominant wind direction is from E to ENE. The months of June and October were considered to be transition months since they did not show a well-defined direction.

*5.2. Ash Cloud Monitoring with Satellite Images*

Due to the large area that a volcanic cloud can cover, satellite images have become a great tool in the monitoring and follow-up of this type of event. The images obtained of Popocatepetl's eruptions reported by the Washington VAAC during the period from 2000 to 2021, show that areas with a higher probability of ash concentration were detected. Based on these regions on a map with lower or higher probability (0.1) of ash was defined. The frequency of occurrence or appearance of ash for 920 MODIS images leads to definition of a concept of frequency of ash appearance in the same region of a map that we call a probability map.

Attempts were later made to identify most of these volcanic cloud events in the MODIS images using the split window technique developed by Prata [36,37]. The split window technique is based on the optical properties of the volcanic ash particles' high silica content, which have an absorption window between 8–13 μm with a peak between 9 and 10 μm [36,38,39]. This allows the presence of volcanic ash clouds to be identified, differentiating these from meteorological clouds composed of water and/or ice droplets. It is important to consider that detection with this type of satellite remote sensing techniques depends on the thermal contrast between the surface and the ash cloud-top temperature in a semi-transparent ash cloud found in the pixel area, which in the case of MODIS is 1 km × 1 km.

In this study, images from NASA's MODIS-Terra and MODIS-Aqua instruments were used. The sensors measure 36 spectral bands in the visible and infrared regions of the electromagnetic spectrum (0.405–14.385 μm) and acquire data at three spatial resolutions: 250 m, 500 m, and 1000 m. The region of the spectrum used to observe volcanic clouds corresponds to bands 28 (7.3 μm), 29 (8.6 μm), 31 (11 μm), and 32 (12 μm) of the MODIS sensors. These four bands are within the thermal infrared range with spatial resolution of 1 km × 1 km at nadir and are sensitive to several volcanogenic species, as $SO_2$ and ash [40].

The Terra and Aqua satellites, launches in 2000 and 2002 respectably, are on a sun-synchronous, near-polar orbit at 705 km above sea level, with a swath width of 2330 km. Each MODIS sensor captures 288 images a day, passing over the area of interest twice a day—once during the night, between 11:00 and 04:00 local time, and the other during the day, between 11:00 and 15:00 local time. Under ideal conditions, it is possible to obtain four images per day. However, there are occasions when the image cannot be used for the detection of volcanic products due to factors such as: (a) the volcano is not observed inside the image; (b) the meteorological conditions (presence of clouds) do not allow observation of the ash cloud; (c) the image is acquired before the eruptive event occurs.

Each image was analyzed to detect the ash emission signature using the brightness temperature difference (BTD) between bands 31 (11 μm) and 32 (12 μm). The brightness

temperature was obtained by the rearranged version of the Planck function formula of radiative transfer [41]

$$T(\lambda, L) = \frac{1.43879 \times 10^4}{\lambda ln([1.19096 \times 10^8 / \lambda^5 L] + 1)} \tag{1}$$

where $\lambda$ is wavelength in microns, $T$ is brightness temperature and $L$ is radiance.

Theoretical calculations were performed using a semitransparent cloud model based on three assumptions: (1) the shapes of the particles are spherical, (2) the particle size distribution is uniform and monodispersed within each pixel, and (3) the cloud forms a single, well-defined, homogeneous layer in each pixel [41]. The detection of volcanic ash is achieved by exploiting selective absorption in the spectral range of Thermal Infrared (TIR), obtaining negative values in pixels with volcanic ash content, which allows identification of the area covered by the ash cloud.

This procedure results in positive pixel values in meteorological clouds. When using satellite images, it is essential to consider the time at which the eruption occurs and the time at which the image is taken, to consider the dispersion phenomena of the volcanic cloud. This time interval is critical in estimating the speed of displacement of the cloud, as well as the possible distance that the plume could reach from the point of emission before it is diluted enough so that the MODIS image cannot detect it.

An essential piece of information in this study was the ash cloud's reported altitude, obtained from the Washington VAAC reports. In the case of Popocatepetl, the ash cloud's altitude was obtained from the reports that CENAPRED delivers to the Washington VAAC based on the continuous monitoring of web cameras. In cases where the ash cloud can be detected by satellite images, the Washington VAAC calculates the temperature of the upper part of the ash cloud that is captured by the satellite and compares it with the atmospheric soundings released by the radiosonde station closest to the volcano (in this case sounding is from Mexico City). It is considered that the cloud reaches its maximum altitude when it is thermally balanced with the air of the surrounding atmosphere and from this equilibrium zone, it is carried by the prevailing wind.

The analysis of the altitudes of the ash clouds is shown in Figure 5, where it is observed that around 88% of the ash clouds reported by the VAAC in the studied period were located between flight levels FL180 (5500 masl) and FL260 (8000 masl). It is important to note that the upper airspace in the Territory of Mexico is designated above FL200 (6100 masl) and, in this area, commercial aircraft carry out air navigation to travel from one airport to another.

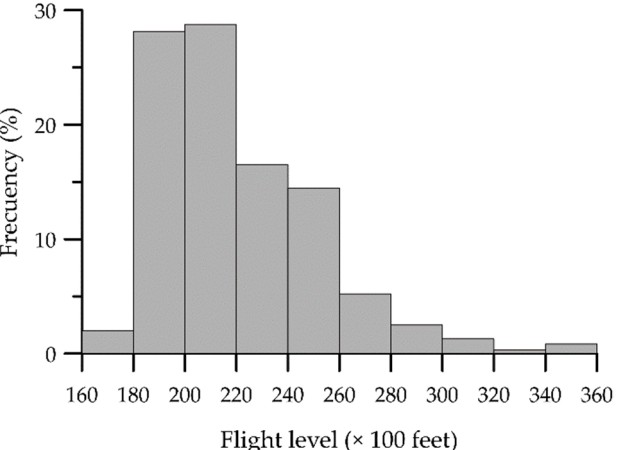

**Figure 5.** Histogram of the altitudes where volcanic ash clouds from Popocatepetl volcano were identified by the Washington VAAC in the period 2000 to 2021. The altitudes are expressed in the nomenclature of flight levels (FL) in hundreds of feet above sea level.



*5.3. Ash Cloud Data Extraction Using MODIS Images*

The combination of the temporal and spatial resolution of MODIS allows a continuous monitoring of the volcano, permitting the identification of more than 62% of the volcanic ash clouds associated with explosive events in the period from 2000 to 2021 (see Table 3). The Washington VAAC in this period reported 1481 eruptive events with ash concentrations of which 920 events were identified in MODIS image. Table 3 shows that, in MODIS images, between 40 and 87% of events were caught each year, which allows a temporary study of the dispersion of ash clouds in the Popocatepetl volcanic area.

**Table 3.** Percentage of Popocatepetl volcano eruptions detected with MODIS images between 2000 and 2021.

| Year | VAAC Eruptions | MODIS Images Match | Eruption Detected in MODIS |
|------|----------------|--------------------|----------------------------|
| 2000 | 59 | 33 | 55.9% |
| 2001 | 49 | 21 | 42.9% |
| 2002 | 12 | 7 | 58.3% |
| 2003 | 35 | 20 | 57.1% |
| 2005 | 21 | 14 | 66.7% |
| 2006 | 13 | 6 | 46.2% |
| 2007 | 8 | 6 | 75.0% |
| 2008 | 8 | 5 | 62.5% |
| 2009 | 9 | 6 | 66.7% |
| 2010 | 5 | 2 | 40.0% |
| 2011 | 14 | 8 | 57.1% |
| 2012 | 54 | 47 | 87.0% |
| 2013 | 52 | 35 | 67.3% |
| 2014 | 19 | 9 | 47.4% |
| 2015 | 80 | 2 | 1.3% |
| 2016 | 79 | 45 | 57.0% |
| 2017 | 89 | 65 | 73.0% |
| 2018 | 140 | 128 | 91.4% |
| 2019 | 262 | 239 | 91.2% |
| 2020 | 286 | 112 | 39.2% |
| 2021 | 187 | 67 | 35.8% |
| Total | 1481 | 920 | 62% (Average) |

The main advantage of detecting volcanic ash clouds utilizing the BTD between the 11 and 12 μm bands, is its fast application and simplistic approach. However, this technique has some drawbacks, such as false alarms ('false positives' when a particular pixel is recognized as ash but does not contain ash and, vice versa, by 'false negatives'), because the radiometric remote sensing techniques of a volcanic ash cloud can be profoundly affected by the presence of dust particles suspended in the atmosphere and the presence of extensive ice content in meteorological clouds and water vapor concentration in the atmospheric conditions that are associated with its geographical location as well as the seasonal variability [42,43]. For this reason, and to identify the area of coverage of the ash clouds around the Popocatepetl volcano, we delimited the search area in the image to a radius of 400 km, considering the maximum distance from the crater to the volcanic ash cloud detected in the MODIS images set already processed.

For each of the images, the pixels corresponding to ash clouds were identified, obtaining data sets characterized by their geographical location. With the matrix of the resulting image, a thresholding process was implemented to segment the bodies within the image (Figure 6a).

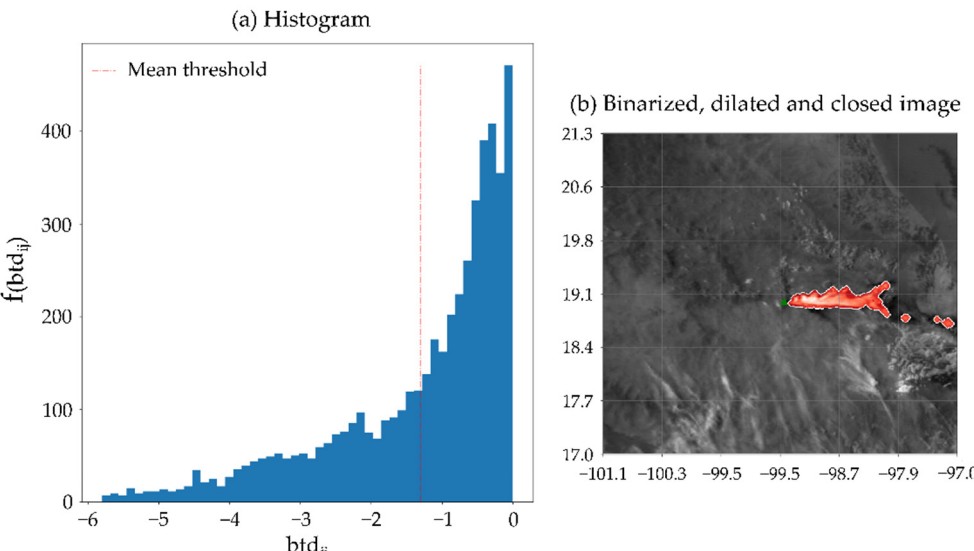

**Figure 6.** Pre-processing of satellite images using media thresholding (**a**) and morphological operations identify an ash cloud with area of 42,000 km$^2$. (**b**) Image corresponding to 20 April 2012 [MODIS Terra, 16:45 UTC].

The BTD method identifies the pixels with ash particles with a negative value and the pixels with the presence of drops of water cloud with positive values [36]. So, by proposing $btd_{ij}$ as the pixel values in the MODIS image, the pixels for $btd_{ij} > 0$ K were discarded. With the pixels for $btd_{ij} < 0$ K a histogram of frequency distribution of events can be expressed as shown in Figure 6a.

A threshold value $T$ was obtained for the pixels that satisfy $btd_{ij} < 0$ K, then the pixels $btd_{ij} \leq T$ were assigned as bodies of volcanic ash. The algorithm selection to compute the threshold $T$ was deduced experimentally, by comparing the results obtained by histogram-based thresholding [44] and automatic-iterative threshold detection [45,46]. The application of the histogram-based mean thresholding achieved the highest number of images correctly binarized, obtaining 178 images with identifiable clouds of volcanic ash, after evaluating a total of 920.

Once the image was binarized, morphological operations were applied over the result to expand and better fit the mask over the truth area (dilation operation), followed by rejecting the noisy pixels (closing operation) [46,47]. This step aims to highlight the found objects' structures and reconstruct them from distorted and noisy shapes, making them more cohesive (Figure 6b). It is important to emphasize that the structure's size for the dilation operation was selected with a small value (4 pixels in diameter), avoiding the attachment of non-ash pixels to main bodies, subsequently processed by the closing operation.

Once evaluating the images was finished, the group of pixels associated with ash was extracted from each image, producing a table in which image label date was stored, along with the pixel's geographical and local matrix image coordinates, and the name of the image. This information can be used in future work as a binary dataset describing the ash and non-ash classes.

With the representative ash pixels obtained from the MODIS image set, the last stage was to identify the areas of the airspace that had the highest percentage of occurrence of volcanic ash from Popocatepetl explosive events in the two periods of the year identified by wind dispersion (from November to June and from July to October). The histogram calculation was made for both geographical variables (latitude and longitude), which allowed the identification of the airspace areas with a high probability of affectation due to the presence of ash in the case of an explosive event.

## 6. Results and Discussion

### 6.1. Wind Patterns

Wind directions are an obvious but non-trivial reason for propagation due to the complex statistical nature of wind distribution over an annual cycle. In general, the wind speed in the lower part of the troposphere is influenced by the area's orography. Therefore, when this influence is lost at higher altitudes, the speeds tend to be higher and more uniform. A statistical analysis of wind speeds cumulative probabilities, shown in Tables 4 and 5, indicates that, during the period from November to May, only 7% of the winds registered at the level of FL180 presented a speed greater than 30 kts, this percentage increasing with increasing altitude reaching a value of almost 54% at the FL300 level. At this higher altitude level in the analysis, maximum speeds of up to 106 kts were presented, but only 14% of the recorded data were above 50 kts. This analysis indicates that the higher the altitude reached by the eruptive column, the greater the speed of the winds responsible for its transport and dispersion, causing the ash cloud coverage area to be more significant.

**Table 4.** Data of the mean, maximum and accumulated frequency of wind speed at levels from 500 to 300 mb in the 20 year period (2000–2020) for the months of November to May.

| Statistical Measures | November–May | | | | |
| --- | --- | --- | --- | --- | --- |
| | 300 mb (FL300) | 350 mb (FL270) | 400 mb (FL240) | 450 mb (FL210) | 500 mb (FL180) |
| Mean: | 32.2 kts | 26.6 kts | 22.4 kts | 17.9 kts | 14.3 kts |
| Maximum: | 106 kts | 98 kts | 98 kts | 76 kts | 72 kts |
| Velocity > 10 kts | 94.6% | 90.5% | 85.3% | 76.2% | 64.3% |
| Velocity > 20 kts | 77.4% | 65.6% | 53.6% | 36.7% | 22.9% |
| Velocity > 30 kts | 54.1% | 38.3% | 24.5% | 13.5% | 7.0% |
| Velocity > 40 kts | 29.8% | 17.4% | 9.5% | 4.2% | 2.2% |
| Velocity > 50 kts | 14.5% | 6.5% | 3.6% | 1.5% | 0.6% |

**Table 5.** Data of the mean, maximum and accumulated frequency of wind speed at levels from 500 to 300 mb in the 20 years period (2000–2020) for the months of July to September.

| Statistical Measures | July–September | | | | |
| --- | --- | --- | --- | --- | --- |
| | 300 mb (FL300) | 350 mb (FL270) | 400 mb (FL240) | 450 mb (FL210) | 500 mb (FL180) |
| Mean: | 11.6 kts | 10.6 kts | 10.3 kts | 10.2 kts | 10.0 kts |
| Maximum: | 40 kts | 39 kts | 38 kts | 40 kts | 42 kts |
| Velocity > 10 kts | 58.3% | 51.5% | 49.9% | 50.9% | 51.8% |
| Velocity > 20 kts | 10.3% | 7.2% | 5.5% | 4.6% | 4.2% |
| Velocity > 30 kts | 0.7% | 0.4% | 0.3% | 0.3% | 0.2% |
| Velocity > 40 kts | 0.0% | 0.0% | 0.0% | 0.0% | 0.0% |
| Velocity > 50 kts | 0.0% | 0.0% | 0.0% | 0.0% | 0.0% |

On the other hand, during the period between July to September, the wind speed at all levels decreased, as shown in Table 5; the maximum wind speeds oscillated around 40 kts at all levels. The data shows that around 50% of the data at all levels of the atmosphere monitored exceeded 10 kts, but the vast majority did not reach 20 kts of speed. This analysis shows that winds at this time of year cause much less dispersion and affectation of ash clouds transported towards the country's center, mainly in directions between SSW and WNW.

### 6.2. Volcanic Ash Detected on Satellite Image

Figure 7 shows four volcanic clouds, for 18 April and 4 May 2012, and 7 March and 9 July 2013. The eruptive event of 18 April 2012 appeared on the MODIS image 5 h after the eruption started presenting an ash cloud covering an area of 2676 km$^2$, moving towards the east, at a speed of 25 kts, as reported by the Washington Volcanic Ash Advisory Center

(VAAC). The ash cloud from the 4 May event was captured by the MODIS sensor 13 h after the eruption began. It covered an area of 3500 km$^2$ airspace, moving WSW at a speed of 10 kts. In the 7 March 2013 event, an ash cloud was detected 5 h after the start of the eruption, which presented an airspace coverage of 4550 km$^2$ in the easterly direction from the volcano's crater, moving at a speed of 20 kts. In the case of the 9 July event, the MODIS image shows the ash cloud 5 h after the beginning of the eruption, which presents airspace coverage of 3300 km$^2$, moving towards the SW at a speed of 10 kts. The four ash clouds were reported at an altitude of around 7000 m (FL230). These observations show that the airspace area that a volcanic cloud can cover depends on three main factors: the altitude of the eruptive column, the wind's speed that moves the cloud, and the continuous emission of ash over long periods.

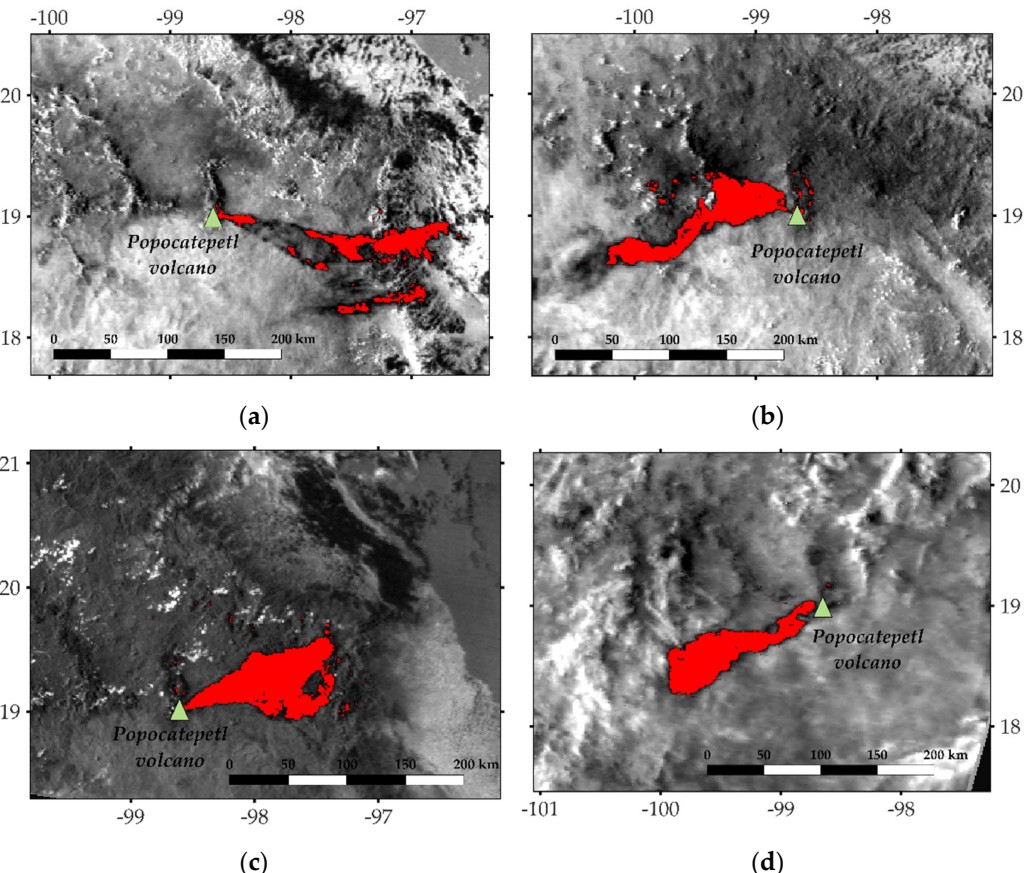

**Figure 7.** Volcanic ash clouds from Popocatepetl volcano identified in the MODIS image by BTD technique. Pixels with negative values that indicate the presence of volcanic ash are identified in red. (**a**) 18 April 2012 [Terra, 16:55 UTC]. Volcanic ash cloud 5 h after the eruption started. (**b**) 4 May 2012 [Terra, 16:55 UTC]. Volcanic ash cloud 13 h after the eruption started. (**c**) 7 March 2013 [Terra, 04:15 UTC]. Volcanic ash cloud 5 h after the eruption started. (**d**) 9 July 2013 [Terra, 17:50 UTC]. Volcanic ash cloud 5 h after the eruption started.

### 6.3. High Frequency Zones of the Presence of Ash

With the data thresholding technique and morphological operations, it was possible to classify 178 MODIS images with well-defined ash clouds, from which geographic information of each pixel representing these clouds was obtained. With this information, occurrence frequency of ash signals in the pixels was calculated for all the images analyzed. This information was separated, taking into account the two periods of the year defined in the wind analysis. With this information, the maps presented in Figure 8 were constructed, defining the areas with highest volcanic ash frequency based on images analyzed around Popocatepetl volcano for different periods of the year.

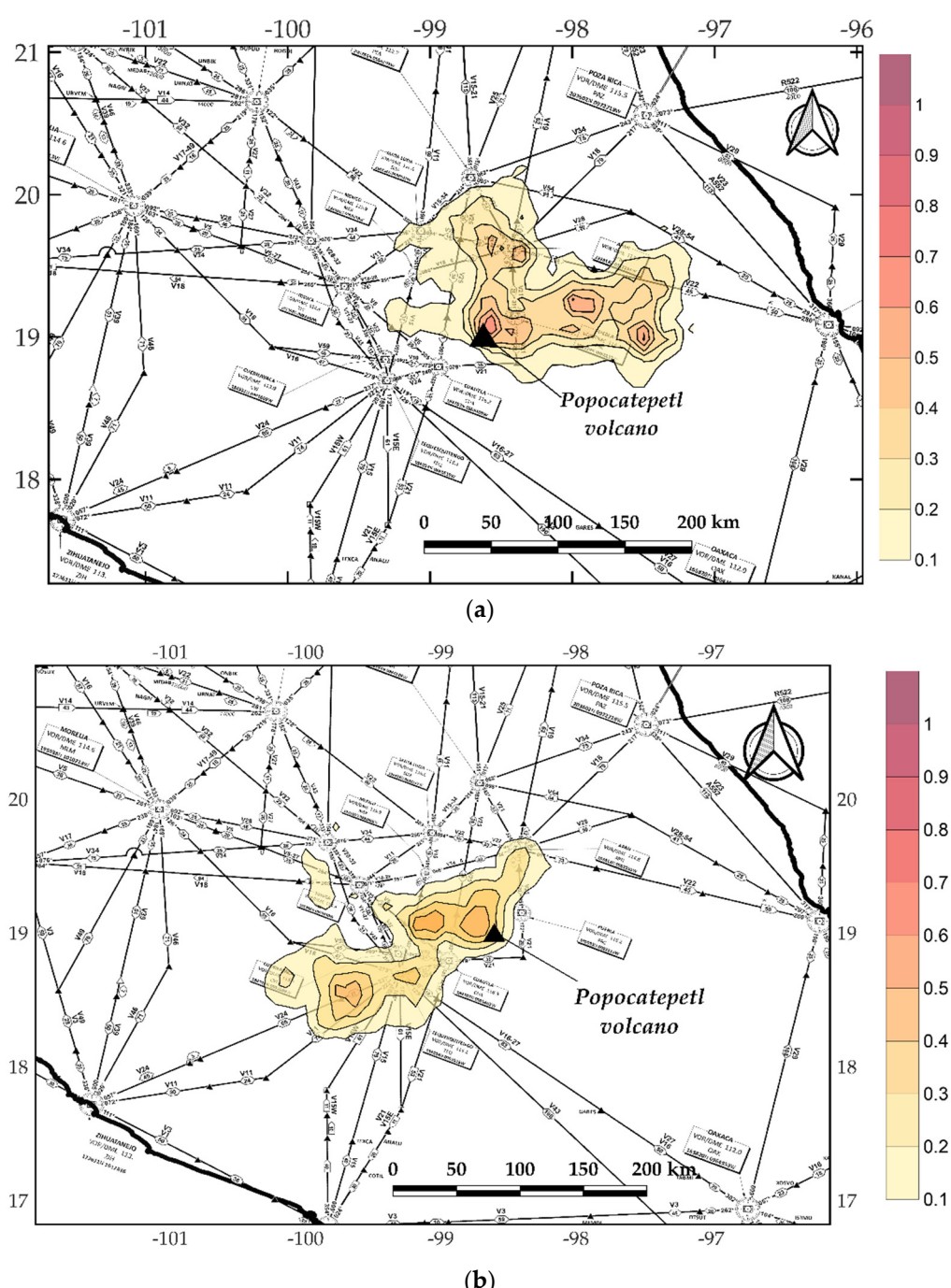

**Figure 8.** Areas with a high probability of being affected by volcanic ash in the event of an eruption of Popocatepetl volcano: (**a**) in the upper part during the November–May period and (**b**) in the lower part during the July–September period.

### 6.4. Aviation Air Routes

To travel from one airport to another, an aircraft must follow pre-established routes controlled by the agencies that regulate the airspace. An airplane can only divert its course with the authorization of the control center that monitors its movement and only in case of an emergency that justifies it. In this work, we used the aeronautical L2 map, published by the Navigation Service offices in the Mexican Air Space SENEAM [48], which is responsible for monitoring and ensuring the safety of aircraft navigation in Mexican territories. This map shows that, within a radius of 200 nautical miles (370 km) around the Popocatepetl volcano, 106 airways are traced that could be affected by ash in an eruption event.

Considering the direction of the winds and the statistics of its speed, reported in the different levels of the atmosphere where data is available, three risk areas were identified, (Figure 9). These areas cover the distance a cloud of ash could reach after one hour of the eruption. As shown in the wind analysis (Section 5.1), the wind has a higher speed in the period November to May. From the observations of the wind speed for this time of the year it was considered that the maximum representative speed was 70 knots whereas 50 knots was used for the rest of the year to estimate the ash cloud's displacement. In this way, the area closest to the volcano's crater represents the area that could be affected during the first hour of the eruption, which can be considered the most critical aviation area after an eruption at Popocatepetl volcano. The second hour after the eruption is identified with the second area's contour, is shown in Figure 9. This illustrates the airspace area that could be affected after the first hour and up to 2 h after the eruption started. A third area indicates an ash cloud location after 2 h but before 4 h after the eruption, taking into account the critical wind speed values.

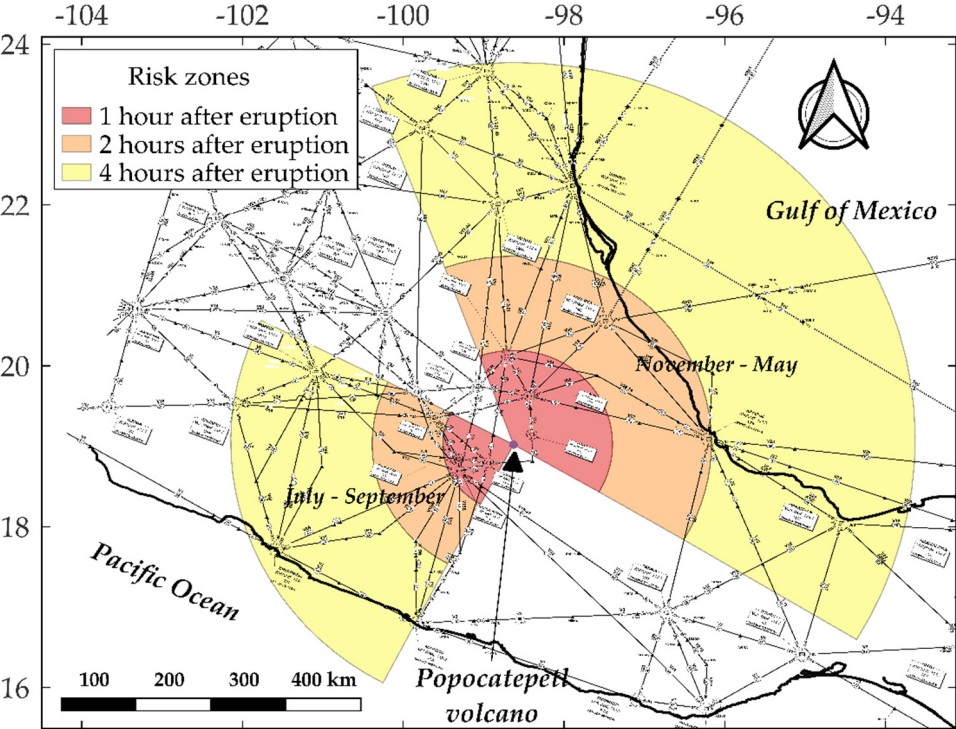

**Figure 9.** Proposed high frequency areas affected by volcanic ash. The map shows delimited areas considering the movement of a volcanic cloud transported by the wind in the seasons of the year defined by the wind direction. Three areas of safety are identified: the first with a radius considering an hour of displacement by wind speed (red); the second level of safety is defined by two hours of transport at a critical wind speed (orange); the third zone by a transport of 4 h at a critical wind speed (yellow).

By considering the data from the wind analysis and the monitoring of ash clouds using satellite images and the numerical model (for a specific period of the year), it is possible to identify the air routes that are most likely to be affected by volcanic ash. In the period November to May, the ash clouds move mainly between NNW and ESE. In this period of the year, ash dispersion can affect the states of Veracruz and Tamaulipas. In this area, aircraft use 36 airways to travel between airports (Figure 9). Between July and September, the area identified as the most likely affected is found on the west side of the volcano between SSW and WNW. The ash clouds that are transported at this time of the year can force the closure of the airspace that links several international airports in the country's central zone, including MCIA, where almost 460,000 operations land and takeoff (reported in official website for 2019 [www.aicm.com.mx], 18 February 2022). In this area

of the country, and due to the lower wind speed, 22 airways are within the area delimited by 4 h of displacement of the cloud in the country's central area, as shown in Figure 9.

## 7. Conclusions

An active volcano represents a high risk to aircraft navigation operations in nearby areas because wind can carry volcanic products, such as ash emitted in an explosive event, depending on its predominant direction and the average wind speed at the level at which it is being transported. Therefore, characterizing the factors that disperse volcanic ash allows us to identify the areas which were most susceptible of being affected by the presence of these volcanic products depending on the time of year when the eruption occurs. This study allows the possibility of identifying zones in the airspace around Popocatepetl volcano with high probability of impact by volcanic ash and this will allow risk mitigation actions in the navigation of aircraft in this zone.

To perform the analysis of the case study, data processing from eruptions of Popocatepetl volcano eruptions over a period from 2000 to 2021 was carried out. It is very important to take into account that the air routes are designed, in essence, for the connectivity of airports and these facilities are built to serve the transportation needs of one or more cities. Popocatepetl volcano is surrounded by important airports and for this reason there are a significant number of airways in its surroundings that are used by both local and international flights. For this reason, it is imperative for risk mitigation due to the presence of volcanic ash in air navigation to understand the nature of the winds in the area. For the identification of the main direction patterns of the plume at different times of the year, a statistical analysis of wind profiles at different levels in the atmosphere where aviation operations prevail was performed. The most frequent wind directions were compared and validated with actual volcanic cloud dispersions observed with MODIS satellite images.

The result of coupling these two tools revealed two main transport patterns for volcanic ash, according to the season of the year. The most extensive period of the year occurs during the months of November to May. It was identified that during these months predominant wind direction occurs between W to SW. This implies that the trend of ash dispersion is mainly between NE to E region. A second, well-defined period occurred during the months of July to September, with predominant wind direction between NNE to E which implies that ash transport was identified with a dominant direction from WSW to W. The months of June and October were identified as transition months, so they do not show a clear trend in atmospheric transport due to the wind. In these months, the dominant wind direction largely depends on the height at which the ash cloud is transported.

From November to May, this period has the highest wind speeds at all altitudes, where up to 15% of winds are >60 knots. During the remaining months (June to October), wind speeds at all altitudes show a significant decrease, with >85% of the data being below 40 knots. This suggests that the ash clouds travel longer distances before they are diluted in the atmosphere.

This information allowed the identification of the zones most likely to be affected by volcanic ash during (and shortly after) explosive events at Popocatepetl volcano, based on the time of the year and for the type of eruptive activity that characterized the volcano in this last eruptive period. In the area defined by 200 nautical miles around Popocatepetl volcano, 106 airways were found to be affected by volcanic eruptions. These airways can be identified, depending on the time of year, as shown on the map presented in Figure 9. From November to May, 36 air routes had a high probability of being affected. Finally, from July to October 22 air routes were identified with a high probability of being affected by volcanic ash in the case of an eruption of Popocatepetl.

The information acquired in this work will allow better planning of alternative routes and airports in the event of an eruption of the Popocatepetl volcano and thus prevent saturation of both air route traffic and airport capacity, helping to reduce the risk of aircraft damage and hence overall safety risk.

**Author Contributions:** All authors contributed to the study conception and design. Conceptualization, J.C.J.-E.; methodology, J.C.J.-E., J.E.A.-R. and R.S.A.-G.; software, R.S.A.-G.; validation, J.L.P.-M., O.M.H.-C. and R.F.D.S.; formal analysis, J.L.P.-M.; investigation, J.R.; writing—original draft preparation, J.C.J.-E.; writing—review and editing, J.R. and R.F.D.S. All authors have read and agreed to the published version of the manuscript.

**Funding:** This research was partially funded by Instituto Politécnico Nacional, under grant SIP-20190186.

**Institutional Review Board Statement:** Not applicable.

**Informed Consent Statement:** Not applicable.

**Data Availability Statement:** The public part of the data base is accessible through: https://ladsweb.modaps.eosdis.nasa.gov/, accessed on 1 January 2022.

**Conflicts of Interest:** The authors declare no conflict of interest.

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
