# Peer review of "Recognition of the Airspace Affected by the Presence of Volcanic Ash from Popocatepetl Volcano Using Historical Satellite Images"

_aerospace, doi:10.3390/aerospace9060308_

Round 1

Author Response

Dear reviewer of the article “Recognition of the airspace affected by the presence of volcanic ash from Popocatepetl volcano using historical satellite images” submitted to the Aerospace Journal. First, we want to thank you for taking the time to review our paper and help us make it better through all your constructive comments.

These observations helped to substantially improve the writing both in content and in figures and tables. The responses to the observations and suggestions made are described in attached document.

Reviewer 2 Report

The manuscript describes the activity carried out to try to identify the areas most frequently subject to the presence of ash following an eruption of the volcano Popocatepetl in Central Mexico. First the authors performed a temporal analysis of the wind direction using NOAA re-analysis data from 1997 to 2016, identifying two different seasonal behaviors (Nov-May; Jul-Sep). Then an analysis of the actual position of the volcanic cloud was carried out by analyzing a series of MODIS images from 2000 to 2021. The ash detection was performed using the consolidated BTD technique, which, however, in some cases can provide false positives and negatives.
This is a statistical approach that can certainly be useful for aviation but which of course is not exhaustive as it does not provide the real position of a volcanic cloud in the event of an eruption.
In my opinion the conclusions are a bit unclear: figure 8 shows the ash probability maps obtained by MODIS while figure 9 shows the hazard area obtained from the wind analysis (the latter is much larger than the first one).
Which of the two is better for aviation to use in the event of an eruption?

Specific Comments:

- 136: 200NM ?? please specify that "NM" means nautical miles
- 169-171: please rephrase this sentence that results unclear
- Fig. 3: in one of the box there's "BDT" instead of "BTD"
- 206: "and the associated gases temperature" ?? unclear
- 212-213: this part must be improved. The NCEP archive must be cited explicitly(https://psl.noaa.gov/data/reanalysis/reanalysis.shtml is this one you used?). What is the spatial resolution of the data?
2.5 deg x 2.5 deg? Which area around the volcano did you choose? Which data were used? 4-times daily, daily or monthly values?
- Table 2: why is there no correspondence between the FL levels used here and in fig.4? Please use the same levels
- Figu 4: the wind direction labels are illegible, please make them bigger. What does the length of the single histograms indicate? The number of cases? The frequency? Please clarify and modify the plot accordingly.
- 224: "550 mb to 300 mb" is wrong, please correct
- 229: please rephrase this sentence that results unclear
- 246: the bands used are four, add here that the spatial resolution of the TIR bands is 1 km x 1km at nadir
- 259-260: please clarify this important aspect, how was this model used? Usually the brightness temperature is obtained simply from radiance using the inverse of Planck formula ...
- 311: what is the "maximum longitude detectable" ?? Please clarify ...
- 319: why you used the symbol "pi_ij" to indicate BTD pixel values? In my opinion can create confusion ... maybe you can use "btd_ij"? 
- 323: what T threshold value was used? If the T threshold value is different image by image, please insert the range (min, max) in which it varies considering all the images.
- 328: correct "identificable" 
- 332: correct "media"
- Fig. 6a: please enlarge the labels. In my opinion "BTD" is better to be used for xlabel. What y-values are? Probability distribution or number of pixels? The values (from 0 to 400) seem to indicate the latter one ...
- 358: "550 mb to 300 mb" is wrong, please correct
- Table 4-5: why here is "300-500 mb", while in fig.4 you used "150-400 mb"? Please uniform these data
- 393: this is "table 5" not "table 4"
- 393: these are the "maximum" wind speeds
- 403-405: this part is better in 5.2 section, not here
- 420: what does it mean "in length"?? Is this an area, right?
- 423: NW?? From fig. 7b seem W ...
- Fig. 7: please if possible use the same grid area for the 4 images. What's in figure is shown? Is it BTD? Please clarify. Text and scale in figures are a bit difficult to read ... maybe you can change colours? In top-right plate text is missing.
- 444: this phrase is unnecessary in my opinion
- fig. 8: please insert a colorbar to identify the values associated to the different colors
- 469: "its speed" ?
- fig. 9: please use a scale in km as the images before. This map was obtained only from wind analysis or also MODIS data were used? Please clarify
- 487: Second level of safety is 2 hours, right?

Author Response

(The authors gave the same response as above.)

Author Response

(The authors gave the same response as above.)

Round 2

Reviewer 1 Report

Thank you for incorporating the suggestions made in my review. I believe the paper is now up to a publishable standard.

Author Response

Dear reviewer of the article, we want to thank you for taking the time to review our paper and help us make it better through all your constructive comments.

Reviewer 2 Report

In my opinion only 3 small adjustments need to be made:

1) It's not yet clear the exact method to detect ash pixels in MODIS images with BTD technique.
In section 4 the authors say:
"The thresholding technique is used after dropping off to zero the pixels with positive values, keeping the negative values as possible ash pixels according to the split window. Therefore, the discrimination threshold is the mean value over all the negative pixels on the image. Then, the values below the mean threshold are taken as ash pixels, being placed commonly around -0.2 K".
In section 5.3 the method is described in more details but it seems a bit different from the one described above: why here you use the histogram with probability distribution? why in fig. 6a the mean threshold is about T = -1.3K (quite different from -0.2K you said before)?
What does it mean "double thresholding process" (line 696)? 

2) Please can you add some more details about how the morphological operations (dilation and closing) were performed (lines 716-746)? What method was used?

3) Please add in the caption of fig.7 (and in fig. 6b too) the detailed description for the MODIS images shown (Terra or Aqua? Time of acquisition (UTC)?). Explicit also what is displayed: is BTD? is Brightness Temperature of channel ... ? Is TOA radiance? It would be better to include a color ramp (grey scale) as well. Correct caption fig. 7c (March 7, not May). Caption fig. 7d: is it July 8 or July 9 as said in line 926?

Author Response

Dear reviewer, we want to thank you for taking the time to review our paper and help us make it better through all your constructive comments on specific points.

Reviewer 3 Report

I appreciate the author's attention to my comments and suggestions, and recommend that this revision be accepted for publication.

Author Response

(The authors gave the same response as above.)
